# SaMoye: Zero-shot Singing Voice Conversion Model Based on Feature Disentanglement and Enhancement

## Abstract

Singing voice conversion (SVC) aims to convert a singer's voice to another singer's from a reference audio while keeping the original semantics. However, existing SVC methods can hardly perform zero-shot due to incomplete feature disentanglement or dependence on the speaker look-up table. We propose the first open-source high-quality zero-shot SVC model SaMoye that can convert singing to human and non-human timbre. SaMoye disentangles the singing voice's features into content, timbre, and pitch features, where we combine multiple ASR models and compress the content features to reduce timbre leakages. Besides, we enhance the timbre features by unfreezing the speaker encoder and mixing the speaker embedding with top-3 similar speakers. We also establish an unparalleled large-scale dataset to guarantee zero-shot performance, which comprises more than 1,815 hours of pure singing voice and 6,367 speakers. We conduct objective and subjective experiments to find that SaMoye outperforms other models in zero-shot SVC tasks even under extreme conditions like converting singing to animals' timbre.

## 1 Introduction

SVC aims to convert the timbre in a given song to the reference audio without disrupting the original content. This technique has wide applications such as virtual singers Nakano & Goto (2011); Hong et al. (2023); Kaewtip et al. (2019), music production Turk et al. (2009); Ijiga et al. (2024) and other artistic domains which currently experience considerable growth thanks to the advances in AI such as AI-based music and art Wang et al. (2022; 2023b; 2024); Wu et al. (2023); Mao et al. (2023). In the SVC task, most singing contents and required timbres are unseen data, which brings huge challenges to SVC models. In recent years, researchers have investigated a variety of approaches to SVC. These studies disentangle the audio into timbre and content features Liu et al. (2021b); Guo et al. (2022); Zhang et al. (2022b). The content features are fused with timbre features from the reference audio and are input into the decoder to generate the audio with timbre converted.

Zero-shot SVC requires the model to perform SVC from short reference audio without finetuning. However, existing SVC models still need minutes to hours of singing clips to fine-tune the model for high-quality singing voice conversion, which vastly constricts these models' usageHuang et al. (2023). There are three possible reasons for this limitation. First, Some studies introduce a speaker lookup table for timbre features Fernandez-Martín et al. (2024); Qian et al. (2020), making it hard to expand the table for unseen speakers. Other studies drop the speaker lookup table and train a speaker encoder to extract the speaker embedding from a given audio. But they confront the second problem called *timbre leakage* in content features. The content features are extracted by pretrained automatic speech recognition (ASR) models like Hubert Hsu et al. (2021), Whisper Radford et al. (2023), or ContentVec Qian et al. (2022). Despite those ASR models keeping the semantic information while reducing the timbre information in the content features, the timbre leakage can

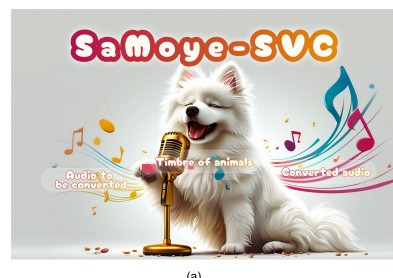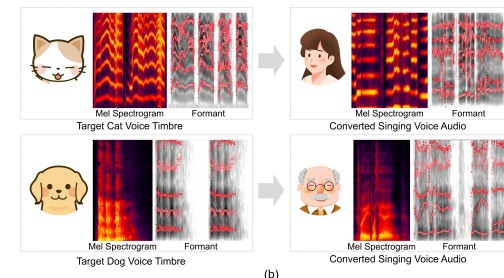

Figure 1: Figure (a) illustrates the functional demonstration of the SaMoye model, which possesses zero-shot sing voice conversion capability to transform both human and non-human timbres like those of cats or dogs. Figure (b) is a specific case analysis of the audio after SaMoye model's conversion, where one can observe that the mel-spectrogram and formant of the converted audio closely resemble those of the animal timbre.

result in the converted audio being more similar to the original instead of reference audio when facing the unseen data. The last reason may be the limited training data. SVC data can be classified as parallel and unparalleled data on whether multiple singers sing the same song. Zero-shot SVC requires the model to see as much data as possible to enhance its generalizing capability. Due to the limited number of parallel data, existing models are normally trained using unparalleled data. However, there is still a low quantity of data for these models to perform zero-shot SVC.

We establish a large-scale unparalleled SVC dataset including 1,815 hours of pure songs with 6,367 speakers by collecting online audio and open-source datasets, and we manually check the datasets to ensure their quality. Besides, we base on Whisper-VITS-SVC[1] to modify the feature disentanglement and inference strategy and propose a zero-shot high-quality SVC model SaMoye can convert human and even non-human timbres. We argued that compressing the content features may improve the timbre leakage and we take multiple approaches to improve the feature disentanglement. Specifically, we evaluate the different combinations of pretrained ASR models including HubertSoft Hsu et al. (2021), Whisper Li et al. (2021), and ContentVec Qian et al. (2022), where we compress the features using vector quantization or k-means. Instead of freezing the speaker encoder in training, we train the speaker encoder together to enhance the timbre information in the speaker embedding. We evaluate SaMoye through objective and subjective experiments on the timbre of humans. As shown in Figure 1, we also assess SVC models on non-human timbres like cats and dogs as the extreme condition, for non-human timbres are absolute unseen data and are effective in assessing the zero-shot ability. The results show that SaMoye can perform SVC with high quality on human and non-human timbre.

The codes and demos for SaMoye are available in the Supplementary_Material_SaMoye. In the demo audios, the first half provides the target reference timbre, while the second half features the converted audio.

To summarize, our contribution includes:

- We establish a large-scale open-source dataset on SVC and propose the first open-source high-quality zero-shot SVC model SaMoye that can even perform SVC for non-human timbre.

- We designed and evaluated the effect of different content features by combining existing ASR models and compression methods like vector quantization and feature clustering.

- We conduct objective and subjective experiments on converting songs to human and non-human timbre to prove that SaMoye can perform high-quality zero-shot SVC.

---

[1]https://github.com/PlayVoice/whisper-vits-svc

## 2 RELATED WORK

### 2.1 SINGING VOICE CONVERSION

The core objective of SVC is to convert the singing voice of a source singer into that of a target singer while preserving the musical content, including melody and rhythm. SVC methods predominantly rely on the recognition-synthesis framework, which involves recognizing the content of the singing voice and then synthesizing it in the target voice. Nercessian (2021) use a pretrained LSTM to extract the speaker embedding and concatenate it with the original speaker's phoneme and loudness embedding in the decoder to generate the converted audio. PitchNet Deng et al. (2020) introduces a singer prediction network and pitch regression network to control the timbre and pitch stability. The embeddings from the two networks are fed into the decoder with the output from the encoder to generate the audio. Luo et al. (2020) use separate encoders to extract singer and techniques embedding for singer and techniques classification tasks respectively and are concatenated before feeding into the decoder and refinement network to generate the converted audio. Polyak et al. (2020) takes as input the speech features by a pre-trained automatic speech recognition (ASR) model Wav2Letter, the F0 feature by Crepe, and the loudness feature from the power spectrum. The speaker embedding from the target singer is included in the generator for converted audio generation. Li et al. (2021) introduce F0 features, PPGs features as content features, and the Mel-spectrogram as the timbre features, which is enhanced through singer classification and reconstruction. FastSVC Liu et al. (2021c) leverages sine-excitation signals and loudness features and uses a Conformer model to extract content features. These studies introduce HiFi-GAN Kong et al. (2020) and BigVGAN gil Lee et al. (2023) as the vocoder to generate the converted audio.

### 2.2 SINGING FEATURE DISENTANGLEMENT

Feature disentanglement is a crucial technique in SVC, as it separates different aspects of the singing voice, such as content, timbre, and pitch, into distinct representations. This separation is essential for achieving flexible and accurate voice conversion. However, few studies have focused on more complete feature disentanglement. Most studies directly apply Hubert Hsu et al. (2021), Whisper Qian et al. (2022), or ContentVec Qian et al. (2022) to extract content features without modification. These models are trained on ASR tasks, resulting in more content and less timbre information in their embeddings. Some studies like Whisper-VITS-SVC introduce auxiliary tasks like speaker recognition to enhance the timbre features. However, few studies have focused on reducing the timbre leakage in the content features. In this paper, we evaluate different content features and apply compression methods like k-means and vector quantization to reduce timbre leakages.

## 3 METHODS

### 3.1 OVERVIEW OF THE SAMOYE

We base on Whisper-VITS-SVC and propose a zero-shot SVC model SaMoye shown in Figure 2. Considering the importance of pitch in singing, we use RMVPE Wei et al. (2023) to extract pitch features. We use GE2E Wan et al. (2020) as the speaker encoder to extract speaker embedding from the audio as the timbre features. For content features, we evaluate various compression methods and combinations of existing ASR models, which will be detailed in Section 3.2. We get the prior distribution from these features using a Flow model, while the posterior distribution is obtained from the original waveform and the speaker embedding using another posterior encoder. SaMoye is trained on audio reconstruction from the posterior distribution aligned with the prior distribution by minimizing their KL divergence. We also introduce a multi-scale discriminator for adversarial learning. In the inference stage, we extract content features, pitch features from

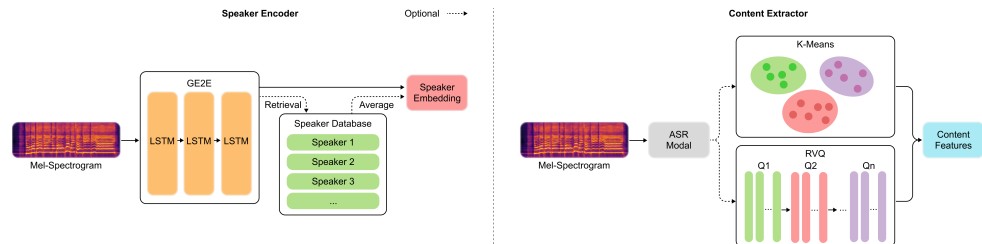

Figure 2: The overall framework of SaMoye. $u_q$ and $log_q$ are of the posterior distribution, and $u_p$ and $log_p$ are of the prior distribution. $z_q$, $z_p$, $z_t$, $z_t$ are sampled from their corresponding latent space, among which $z_t$ is from the forward-process of the Flow model and $z_f$ is from the inverse process.

Figure 3: The process of extracting the speaker embedding (left panel) and content features (right panel). We applied at least one of the pre-trained models for the ASR model, including HubertSoft, ContentVec, and Whisper.

the original audio, and timbre features from the reference audio to get their prior distribution through the Flow model and generate the converted results.

## 3.2 FEATURE DISENTANGLEMENT AND ENHANCEMENT

We improve feature disentanglement by reducing content feature timbre leakages and enhancing the speaker encoder's timbre information. As is depicted in Figure 3, we evaluate various combinations of HubertSoft, Whisper, and ContentVec using concatenation and try to compress their features using k-means or vector quantization. Specifically, we implement vector quantization by a residual vector quantization through a set of codebooks. For timbre features, we unfreeze the speaker encoder during training.

## 3.3 SAMOYE (RETRIEVAL)

We are inspired by retrieval-based voice conversion to introduce another inference strategy, namely SaMoye (retrieval). SaMoye (retrieval) keeps the same architecture as SaMoye but retrieves the top-3 most similar speaker embeddings from the speaker embedding database and averages them with the reference speaker embedding in the inference stage. We build the speaker embedding database from the dataset to compute the cosine similarity between the reference audio and the embedding in the database. Then, in the inference stage, we can retrieve the top three similar speaker embeddings and average them with the reference speaker embedding as the timbre features; this method may reduce the timbre similarity of the converted audio but

improve the zero-shot performance by exploiting the seen speaker embedding to fill the gap brought by the unseen reference audio that can result in bad cases like mute voice.

### 3.4 ADVERSARIAL LEARNING

We follow Whisper-VITS-SVC and introduce an adversarial training strategy to train the generator to synthesize the converted audio and a discriminator to discriminate between the generated and real audio. For the generator, the Mel-spectrogram and speaker embedding pass through the posterior encoder to get the posterior latent variables $z_q$. We apply the decoder in gil Lee et al. (2023) to generate the audio from the latent variables. Meanwhile, the pitch and content features pass another encoder to get the latent variables $z_p$. Inspired by gil Lee et al. (2023), we use the speaker embedding as the condition for a Flow model to generate the prior latent variables $z_t$ to be aligned with the posterior latent space. We compute the Kullback–Leibler divergence(KL) between the $\mu$ and variance $\sigma^2$ of $z_t$ and $z_q$ as well as $z_f$ and $z_p$ to sum up as the $\mathcal{L}_{kl}$. We also sum up the *L1* and *L2* loss for waveform as $\mathcal{L}_{wav}$ and the mel-spectrogram as $\mathcal{L}_{mel}$ for audio reconstruction. We also use the $\mathcal{L}_{stft}$ following Takaki et al. (2019).

$$\mathcal{L}_{dis} = E\left[D(a)^2 + (1 - D(a'))^2\right] \tag{1}$$

$$\mathcal{L}_{adv} = E\left[(1 - D(a'))^2\right] \tag{2}$$

For the discriminator, we use the multi-scale and multi-period discriminator in our study, which takes the generated and real audio as input to compute the *L1* loss between their feature maps in the discriminator as $\mathcal{L}_{fmap}$. The loss for the discriminator is shown in equation 1 and the adversarial loss for the generator is described in equation 2, where $D$ is the discriminator and $a$ and $a'$ is real and generated audio.

The final loss for the generator is shown below, where we set the $\alpha$ to 1.0, $\beta$ to 0.2, and $\gamma$ to 9.0 during training.

$$\begin{aligned}\mathcal{L}_{gen} = \mathcal{L}_{wav} + \mathcal{L}_{mel} + \beta * \mathcal{L}_{kl} \\ + \mathcal{L}_{adv} + \mathcal{L}_{fmap} + \gamma * \mathcal{L}_{stft}\end{aligned} \tag{3}$$

## 4 EXPERIMENT

### 4.1 DATASETS

We have utilized a dataset comprising 6,367 speakers and over 1,815 hours of data for training purposes. This dataset includes 36.5 hours of online music and over 1,815 hours of open-source singing and speech data. We separate the pure human voice from these music clips using Demucs Défossez et al. (2019) for online music. Consequently, the obtained data are manually checked to remove all the errors in the recognition results. The specifics of these datasets are detailed in Table 1.

### 4.2 EXPERIMENT SETUP

The models are trained on the established datasets. All the audio sampling rates are united to 32k. We use a filter length of 1024, a hop length of 320, and a Hanning window with a length of 1024 when converting the waveform to the Mel-spectrogram with 80 filters.

We select five song clips with a length of 20 seconds and 11 speakers (6 humans, and 5 non-humans) to evaluate the models' performance on zero-shot SVC including the extreme condition for non-human timbre.

Table 1: The statistics of the large-scale open-source dataset which is used in SaMoye-SVC.

| Datasets | Speakers | Duration(hours) |
|---|---|---|
| JSUT-SongSonobe et al. (2017) | 1 | 0.41 |
| PJSKoguchi & Takamichi (2020) | 1 | 0.60 |
| KiSingShi et al. (2022) | 1 | 0.88 |
| Jvs MusicTamaru et al. (2020) | 100 | 4.00 |
| CSDChoi et al. (2020) | 1 | 4.86 |
| OpencpopHuang et al. (2021) | 1 | 5.20 |
| DSD100Liutkus et al. (2017) | 100 | 6.99 |
| PopcsLiu et al. (2021a) | 117 | 5.89 |

| Datasets | Speakers | Duration(hours) |
|---|---|---|
| KSS Park (2018) | 1 | 12.85 |
| Online Music | 203 | 36.5 |
| M4SingerZhang et al. (2022a) | 20 | 29.77 |
| OpenSingerHuang et al. (2021) | 66 | 50.00 |
| VCTK Valentini-Botinhao et al. (2017) | 109 | 44.00 |
| Aishell-3Shi et al. (2020) | 218 | 85.00 |
| DAMP VPBSmule (2017) | 5428 | 1529.00 |
| Total | 6367 | 1815.95 |

The five songs cover the full range from bass to treble. The human timbres contain 4 males and 2 females, while the 5 non-humans include 3 cats and 2 dogs.

For the subjective experiment, we invite 20 music-professional participants to listen to the original music and timbre and evaluate the converted song clips.

## 4.3 METRICS

We use several objective and subjective metrics to evaluate the quality of the converted audio and how similar the timbre is to the original audio.

### 4.3.1 SUBJECTIVE METRICS

The subjective metrics include:

- Mean Opinion Score on Similarity(MOS-S): MOS-S is based on a 5-score Likert scale, where 5 means the same timbre and 1 for a completely different timbre.

- Mean Opinion Score on Quality(MOS-Q): MOS is a widely-used audio or video quality evaluation standard based on expert evaluation. The score of MOS is from 1 to 5, where a higher score means higher quality.

### 4.3.2 OBJECTIVE METRICS

The following various objective metrics are also included to evaluate the converted audio from different perspectives:

- Perceptual Evaluation of Speech Quality (PESQ): PESQ computes multiple perspectives like temporal alignment and perceptual filtering. PESQ score is from -0.5 to 4.5, where a higher score stands for better perceptual quality.

- Short-Time Objective Intelligibility (STOI): STOI represents how well the audio can be comprehended, and the STOI score is from 0 to 1. The higher STOI score means that the audio is easier to understand.

- Non-Intrusive Speech Quality Assessment (NISQA) [2]: NISQA is a non-intrusive metrics based on a pre-trained deep learning model. Given audio, the NISQA predicts MOS for audio quality, NOI for noise degree, DIS for audio coherence, COL for timbre quality, and Loud for loudness. All these predicted scores are the higher the better.

---

[2]https://github.com/gabrielmittag/NISQA

Table 2: Objective and subjective results of SaMoye and baseline models **(Human)**.(All subjective metrics exhibit statistically significant differences with p < 0.03.)

| Model | Objective Metrics | | | | Subjective Metrics | |
|---|---|---|---|---|---|---|
| | PESQ↑ | STOI↑ | NISQA↑ | SECS↑ | MOS-S↑ | MOS-Q↑ |
| Sovits-SVC | 1.818 | 0.572 | 2.893 | 0.257 | 2.98 | 3.50 |
| FreeVC | 0.200 | 0.466 | **3.807** | 0.236 | 1.36 | 1.25 |
| GPT-Sovits | 0.332 | 0.447 | 1.755 | 0.201 | 1.52 | 1.28 |
| SaMoye (retrieval) | 2.053 | 0.622 | 3.193 | 0.192 | **3.24** | **3.80** |
| SaMoye | **2.343** | **0.640** | 2.278 | **0.266** | 3.17 | 3.64 |

Table 3: Objective and subjective results of SaMoye and baseline models **(Non-human)**.(All subjective metrics exhibit statistically significant differences with p < 0.05.)

| Model | Objective Metrics | | | | Subjective Metrics | |
|---|---|---|---|---|---|---|
| | PESQ↑ | STOI↑ | NISQA↑ | SECS↑ | MOS-S↑ | MOS-Q↑ |
| Sovits-SVC | 2.085 | 0.608 | 3.056 | 0.208 | 2.80 | 3.47 |
| FreeVC | 0.269 | 0.475 | **4.112** | 0.178 | 1.25 | 1.27 |
| GPT-Sovits | 0.642 | 0.454 | 1.507 | **0.356** | 1.45 | 1.24 |
| SaMoye (retrieval) | 2.057 | 0.627 | 3.221 | 0.164 | **3.09** | **3.88** |
| SaMoye | **2.392** | **0.652** | 2.762 | 0.241 | 2.95 | 3.47 |

- Speaker Encoder Cosine Similarity (SECS): We use CAM++ Wang et al. (2023a) to extract speaker embedding from generated and original audio and compute their cosine similarity. SECS score is from 0 to 1, and higher SECS means higher timbre similarity.

## 4.4 COMPARISON

We compare SaMoye with several SVC or VC models to evaluate its performance. These models are trained on our datasets and their details are as follows:

- SoVITS-SVC[3]: The flow model based Sovits-SVC, which uses *Contentvec* as its representations.
- FreeVC Li et al. (2023): FreeVC is a speech voice-conversion model, which disentangles content information by imposing an information bottleneck to WavLM features, and introduces the spectrogram-resize-based data augmentation to improve the purity of extracted content information.s
- GPT-SoVITs[4]: GPT-SoVITs is originally a TTS-model. We transfer GPT-SoVITS to singing voice conversion by replacing the phoneme embedding in GPT-SoVITS with F0 embedding and training on our datasets.

The results are shown in Table 2. SaMoye outperforms other models in PESQ, STOI, and SECS. We also noticed that FreeVC achieves the highest NISQA. The reason may be that NISQA is evaluated by a model trained on speech datasets to predict MOS scores, while there is a gap between singing and speech. FreeVC is a voice-conversion model, in which the converted audio sounds like speech instead of singing. For subjective

---

[3]https://github.com/svc-develop-team/so-vits-svc
[4]https://github.com/RVC-Boss/GPT-SoVITS

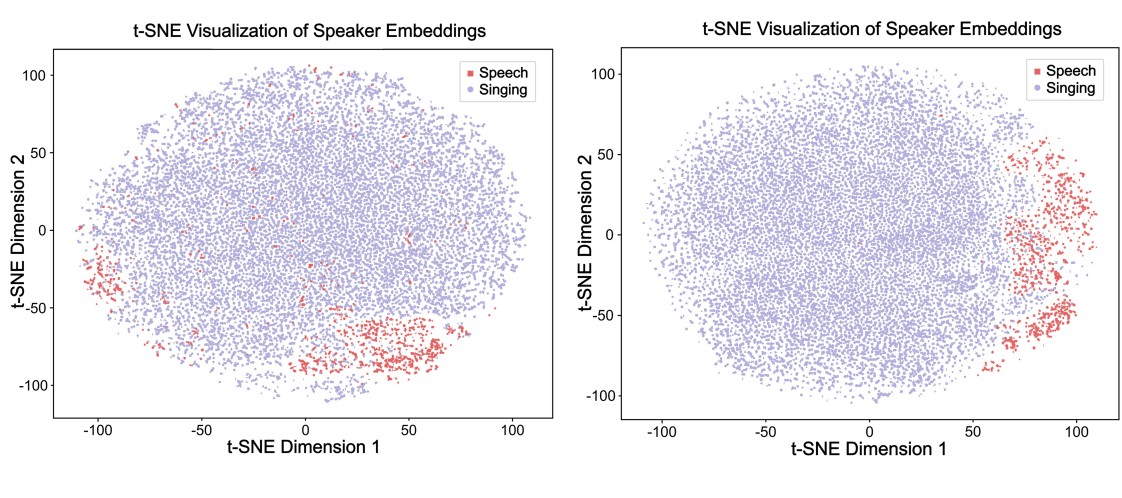

(a) Fixed speaker encoder.              (b) Unfrozen speaker encoder.

Figure 4: The t-SNE visualization of the speaker embedding from fixed (left) and unfrozen (right) speaker encoders. Each point represents a speaker with a total of 6299 speech and 329 singing audios are selected.

evaluation, we observed that SaMoye (retrieval) performs the best in both MOS-S and MOS-Q because SaMoye (retrieval) mixes the top three similar speaker embeddings, which fill the information gap in contrast with just using single speaker embedding; this is proved through the lowest SECS for SaMoye (retrieval) because the mixture impact some of the timbre features extracted from the reference audio. Furthermore, we conclude that audio quality can significantly influence the timbre similarity, for models with low MOS-Q can hardly achieve MOS-S. Despite the speaker embedding difference introduced by SaMoye (retrieval), listeners are insensitive to the influence when the audio quality is high.

### 4.4.1 EVALUATION ON ANIMAL TIMBRES

We use the animal voice as the extreme condition to perform zero-shot singing voice conversion for they are complete unseen data for all SVC models. The results are shown in Table 3. The results are nearly the same as the comparison experiment. However, we find that GPT-Sovits achieves the highest SECS. This may be attributed to that the animal voice is still unseen data for CAM++ which is trained on human speech datasets. Because of the inaccuracy brought by these unseen data, the SECS is less reliable under this extreme condition. This is proved by the subjective metrics, where GPT-Sovits results in low MOS-S and MOS-Q scores. The results further explain the conclusion in the human voice cases that SaMoye (retrieval) can fill the blank information in the speaker embedding, especially for the animal cases where the timbre information blank is huger.

### 4.5 EVALUATION ON FEATURE DISENTANGLEMENT

We investigate the most used content feature extractors including HubertSoft, Whisper, and ContentVec, all trained on ASR tasks. We also evaluate the features from the different layers of ContentVec. Besides, we introduce different compression methods including k-means and RVQ (Residual Vector Quantization) to reduce the timbre leakage in the content features. We use *Tensor* methods to input the cluster center of

Table 4: Objective results for different feature representations.

| Setting | Model | Objective Metrics | | | |
|---|---|---|---|---|---|
| | | PESQ↑ | STOI↑ | NISQA↑ | SECS↑ |
| #1 | Whisper | 2.073 | 0.613 | 2.833 | 0.270 |
| #2 | HubertSoft | 2.013 | 0.522 | **2.889** | **0.278** |
| #3 | ContentVec (Layers 9) | 1.032 | 0.478 | 2.776 | 0.187 |
| #4 | ContentVec (Layers 12) | 0.957 | 0.483 | 2.620 | 0.205 |
| #5 | HubertSoft+k-means(900, Tensor) | 0.497 | 0.376 | 2.499 | 0.215 |
| #6 | HubertSoft+k-means(900, Codes) | 0.272 | 0.214 | 1.726 | 0.165 |
| #7 | HubertSoft+RVQ(Tensor) | 0.847 | 0.352 | 2.131 | 0.189 |
| #8 | HubertSoft+RVQ(Codes) | 1.071 | 0.230 | 2.483 | **0.278** |
| #9 | HubertSoft+Whisper | **2.343** | **0.640** | 2.278 | 0.266 |

Table 5: The results of fixed and unfrozen speaker encoder. (All subjective metrics exhibit statistically significant differences with $p < 0.05$.)

| Speaker Encoder | Objective Metrics | | | | Subjective Metrics | |
|---|---|---|---|---|---|---|
| | PESQ↑ | STOI↑ | NISQA↑ | SCES↑ | MOS-S↑ | MOS-Q↑ |
| Fixed | 2.105 | 0.614 | **2.796** | 0.250 | 3.05 | 3.35 |
| Unfrozen | **2.343** | **0.640** | 2.278 | **0.266** | **3.17** | **3.63** |

k-means or codebook embeddings into the model, while for *Codes* we use the cluster index for k-means or codebook's index for RVQ instead. The results are shown in Table 4. We noticed that Hubert and Whisper perform better than ContentVec. Setting #2, #7, and #8 prove that HubertSoft contains less timbre leakage of the original audio and results in higher similarity in the converted audio. Setting #6, and #8 that use codes have worse performance than those using embeddings; the reason may be that the codes learned from k-means and RVQ lose too much information, which makes it harder for the model to understand the original content. Considering the overall performances, we use setting #9 as the best model for further experiments.

We also evaluate the effect of unfreezing the speaker encoder during training. The results are shown in Table 5 that unfreezing the speaker encoder during training enhances the performance in all metrics except NISQA, which may be because the NISQA model is pretrained on speech datasets where a gap exists between the singing and speech. In addition, we use t-SNE to reduce the speaker embedding to two dimensions for visualization as shown in Figure 4. We select 328 speakers from speech datasets including Aishell, KSS, and VCTK, and 6,299 speakers in our singing datasets. The results show training the speaker encoder on the SVC tasks makes a clearer difference between the speaker embedding from speech and singing. It also proved that the timbre in singing is more complex and different from speech, which can be caught through the unfrozen speaker encoder.

## 5 CONCLUSION

This paper proposes an open-source large-scale dataset and a high-quality zero-shot SVC model that can convert a song to human and even non-human timbre. We investigate multiple methods for disentangling the content features by combining different ASR models and introducing K-means and vector quantization to compress these features. We also enhance the timbre feature by training the speaker encoder together. We conduct objective and subjective experiments to find that SaMoye can perform high-quality SVC on human and non-human timbre.

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
