# OpenReview forum: "SaMoye: Zero-shot Singing Voice Conversion Model Based on Feature Disentanglement and Enhancement"
_ICLR.cc/2025/Conference — ICLR 2025 Conference Withdrawn Submission_

### Official Review · Reviewer_HpWU · 2024-10-28

**Soundness:** 2
**Presentation:** 2
**Contribution:** 2
**Rating:** 3
**Confidence:** 4

**Summary:**

This paper studies zero-shot sing voice conversion. The authors follow a disentanglement framework to disentangle the singing voice's features into content, timbre, and pitch features.  A combination of multiple ASR models and joint training with speaker encoder are introduced to reduce the timbre leakage and improve the speaker similarity.

**Strengths:**

1. The authors extend the timbre reference from the unseen speaker to animals, which is an interesting exploration.
2. The authors provide a dataset for singing voice conversion investigation.

**Weaknesses:**

1. While the singing voice conversion is an interesting task, the contribution of this paper is limited. The combination of multiple ASR or SSL models to disentangle linguistic content is widely employed in many VC works. The joint training between the backbone model and the pre-trained speaker encoder is also not a new approach. The authors should strengthen their claims by providing a clearer distinction between their work and prior models.
2. Section 3.2 of feature disentanglement and enhancement does not provide sufficient detail. A more thorough explanation of how the combination of content features operates and the details of k-means or vector quantization, would enhance understanding.
3. The experimental results are strange, it makes me confused as to why the PESQ of FreeVC and GPT-Sovits is about 0.2 and 0.3, respectively. Meanwhile, the speaker similarity of all systems is significantly lower than normal values. Please clarify this issue.
4. There are fewer demos to display the performance and the existing demos of animal conversion have an undesirable similarity. It would be nice to provide more audio samples.
5. Some trivial issues:
 - SVC -> Singing Voice Conversion (SVC) in Line 29
 - Miss blank between usage and Huang in Line 40
 - Huber and ContentVec are self-supervised models instead of ASR models

**Questions:**

1. How do we evaluate the similarity of converting the timbre into animals? Current speaker verification models are trained using human speech, is there a mismatch?

---

### Official Review · Reviewer_YbuM · 2024-10-28

**Soundness:** 2
**Presentation:** 2
**Contribution:** 2
**Rating:** 3
**Confidence:** 5

**Summary:**

The paper introduces a novel zero-shot singing voice conversion method. The proposed approach leverages the Flow model in conjunction with speaker embeddings and audio codes, aiming to achieve high-quality voice conversion. The model is further optimized using a GAN-based loss function.

Key contributions of the paper include:
1. The release of an open-source dataset to facilitate future research.
2. A comprehensive evaluation of different content features, employing ASR models, Residual Vector Quantization (RVQ), and k-Nearest Neighbors (KNN).
3. Objective and subjective experiments conducted to assess the performance of the proposed singing voice conversion framework.
4. A novel attempt to extend the model's capabilities by converting non-human timbre into human timbre, addressing a unique challenge in voice synthesis research.

**Strengths:**

1. The paper presents a flow-based architecture designed specifically for singing voice conversion, demonstrating improved performance in transforming vocal timbres while maintaining naturalness and expressiveness.
2. The study introduces a novel method for converting non-human voices into human-like voices, addressing key challenges in voice synthesis and bridging the gap between synthetic and natural speech.
3. The paper releases a new dataset, providing valuable resources for further research in voice conversion and fostering advancements in related fields through the availability of high-quality data.

**Weaknesses:**

1. The audio outputs are not sufficiently convincing, making them challenging to listen to and assess effectively.

2. The concept of singing voice conversion has been extensively studied in the literature, particularly through the use of flow-based and diffusion models. Notable examples include:
Real-Time and Accurate: Zero-shot High-Fidelity Singing Voice Conversion with Multi-Condition Flow Synthesis [https://arxiv.org/abs/2405.15093]

SingVisio: Visual Analytics of Diffusion Model for Singing Voice Conversion [https://arxiv.org/pdf/2402.12660]

3. There is a potential issue of timbre leakage from the automatic speech recognition (ASR) model, which could affect the conversion quality.

**Questions:**

1. **Why use SECS (Speaker Similarity) for non-human audio?** The effectiveness of speaker embeddings for non-human audio remains uncertain, raising questions about their reliability in such contexts.

2. **Incorporate a speaker classification model for enhanced evaluation.** Introducing a speaker classification model would provide a more robust assessment of performance, ensuring a comprehensive evaluation beyond speaker similarity metrics.

---

### Official Review · Reviewer_dwTe · 2024-11-04

**Soundness:** 2
**Presentation:** 1
**Contribution:** 1
**Rating:** 3
**Confidence:** 5

**Summary:**

This paper aims to address the issue of timbre leakage in zero-shot singing voice conversion (SVC). The authors leverage content features obtained from various content-based pretrained models and utilize a mixed speaker embeddings approach to mitigate timbre leakage. They conduct experimental investigations using a 1.8k-hour singing voice corpus, drawing data from academic speech/singing corpora as well as relatively in-the-wild sources like the DAMP singing voice corpus.

**Strengths:**

1. Timbre leakage is indeed a critical issue in the fields of zero-shot VC and SVC. Finding ways to reduce speaker characteristics embedded within pretrained content features is a crucial area requiring further exploration and resolution.

2. The authors have invested considerable effort into experimental investigations of various content-based representations, including both continuous hidden features and discrete codecs.

3. The mixture strategy based on the top-3 similar speaker embeddings is a good engineering trick.

**Weaknesses:**

Overall, I find this paper to be an experimental exploration in the zero-shot SVC task. However, it resembles more of an engineering-focused technical report rather than an innovative research paper. Specifically:

1. Regarding novelty: (1) Existing SVC literature has already explored combining various content-based pretrained models (e.g., Whisper, ContentVec) [1]; (2) Using vector quantization to remove speaker information was first suggested in the VQ-VAE paper [2], where the authors noted its potential application in voice conversion. This approach has also been implemented in works like VQ-VC [3]. K-means clustering, similarly a form of vector quantization, is already a well-known method for addressing timbre leakage in the VC field.

2. Some of the authors' terminology is misleading and lacks rigor. For example: (1) HuBERTSoft and ContentVec should not be labeled as ASR models—they are self-supervised models; (2) The authors claim that "we establish an unparalleled large-scale dataset," but the term "establish" is highly misleading. In reality, the authors mainly aggregated multiple existing datasets for their experiments, with only 36.5 hours of Online Music being newly added in this study; (3) In Table 5, "Fixed" should correspond with "Unfixed," while "Unfrozen" should pair with "Frozen."

3. Despite utilizing numerous engineering tricks, the final audio sample quality remains subpar. The authors should at least compare with a diffusion-based SVC baseline.

4. The emphasis on mimicking the timbre of a dog or cat is not a unique contribution of this paper. Any SVC model can conduct inference with dog or cat samples.

> [1] Leveraging Diverse Semantic-based Audio Pretrained Models for Singing Voice Conversion. IEEE SLT 2024.
>
> [2] Neural discrete representation learning. NIPS 2017.
>
> [3] One-shot voice conversion by vector quantization. ICASSP 2020.

**Questions:**

The authors should strengthen their literature review of the broader speech synthesis field, including VC and TTS, and reconsider the contributions of this study in that context.

---

### Official Review · Reviewer_zYpW · 2024-11-05

**Soundness:** 2
**Presentation:** 1
**Contribution:** 1
**Rating:** 3
**Confidence:** 5

**Summary:**

The author proposes a zero-shot singing voice synthesis model that aims to achieve better feature space disentanglement than the current state-of-the-art. It claims to achieve this goal using the following:

1. A large-scale singing voice dataset including both human and non-human voices.
2. Use of the VITS architecture, along with multiple ASR modules for improved content feature extraction to reduce timbre leakage.
3. Inference-time speaker embedding retrieval using cosine similarity.

**Strengths:**

1. The problem statement is clear.
2. The baseline, with so-VITS and FreeVC, is sufficient.

**Weaknesses:**

3. The theoretical explanation is poor. In particular:

	3.1 This is essentially a re-explanation of the VITS architecture, with no significant novelty found.
	3.2 This section should present the main theoretical novelty of the paper; however, the algorithm, including the use of k-means and quantization, remains unclear.
	Figure 3(a) suggests that the speaker encoders are directly interfacing with the spectrograms, which implies that unfreezing them during training could allow timbre leakage.
	Figure 3(b) is unclear. How is the final vector produced from the clusters and codebooks? Are Q1, Q2, etc., referring to the codebooks? Do you use both methods simultaneously, or only select one?
	3.3 Mapping to the closest speaker embeddings in the training set "may reduce the timbre similarity of the converted audio." However, since one of the main goals stated in the abstract is to avoid timbre leakage, what is the rationale behind using this technique?
	3.4 The loss components presented are not novel, as all of them are already found in the HiFi-GAN and/or FreeVC papers.

	- Regarding the dataset, while it is noted as a novel aspect, this is left unexplained in Section 3.

4. Experiments/VisualizationThe experimental details are lacking. Specifically:
	4.2 Where are the training parameters? The only parameters specified are those in the loss function equation (3). Did you re-train the baselines on the proposed dataset as well? Furthermore, with such a large-scale dataset, why would you only evaluate on 11 speakers for 100 seconds each? For the non-human part, are you testing zero-shot conversion from humans to non-humans, non-humans to humans, or within each group only?

	Tables 2 and 3: Although seemingly complete and acceptable metrics-wise, these contain two issues:
		- Since the primary goal is reducing timbre leakage, why is there only a trivial gain on SECS in both human and non-human cases? For example, in Table 1, your gain above the original so-VITS-SVC is only 0.009 — is that really statistically significant?
		- The retrieval method you proposed has a much lower SECS. How, then, does it achieve the highest MOS? Does that mean a user would rate 5/5 if I presented them with unconverted audio?


	Section 4.1: The Tensor/Codes method and Table 4 do not make sense without adequate explanations in Section 3.2.

	Figure 4: What is the purpose of this figure? In the text, it is mentioned that "The results show that training the speaker encoder on the SVC tasks makes a clearer distinction between the speaker embeddings from speech and singing." But isn't the primary goal to demonstrate improvements in SVC performance?

**Questions:**

--

---

### Official Review · Reviewer_JQTa · 2024-11-06

**Soundness:** 3
**Presentation:** 2
**Contribution:** 2
**Rating:** 3
**Confidence:** 3

**Summary:**

The paper presents SaMoye, a method for singing-voice conversion in the zero-shot setting. The method extends existing approaches for SVC in a few different ways:
1. The method attempts to disentangle timbre, content and pitch by combining features from multiple ASR models
2. The method involves an enhanced set of timbre features by training the speaker encoder in conjunction with the model instead of using a pre-trained and frozen speaker encoder or a speaker embedding table.
3. The timbre features are also enhanced by mixing the top-3 most similar speakers' embeddings.

The paper involves a large collation of singing voice datasets and the authors perform both subjective and objective tests to evaluate their method.

**Strengths:**

The paper's objectives are very clear and the figures do a good job of explaining how the model architecture works. The paper does a thorough evaluation of their proposed ideas in the ablation studies showing the validity of each of their contribution.

**Weaknesses:**

The insights from this paper do not seem very significant for the community at large.
The main idea is to incorporate more data, and add more conditioning variables to disentangle timbre, content and pitch. However, even then, the contributions in the paper seem very narrow. The authors have stated that previous work has utilized various speech encoders to try to disentangle content. Here, they used a combination of encoders instead.

Another complaint is that the paper seems not very polished and needs a round of proofreading. It also seems like the template is different from the ICLR paper template. The bottom margin has a lot of space and the text width seems wider.

**Questions:**

I guess the main question is, how are the findings in the paper significant for the broader community? While it is true that the method proposed involves steps that have not been explored in this specific topic, I am interested in seeing if there are resuable insights for the general audience.

---

### Note · Authors · 2024-11-13

I have read and agree with the venue's withdrawal policy on behalf of myself and my co-authors.